# Analytical sensitivity and clinical performance of "COVID-19 RT-PCR Real TM FAST (CY5) (ATGen, Uruguay) and "ECUGEN SARS-CoV-2 RT-qPCR" (UDLA-STARNEWCORP, Ecuador)": High quality-low cost local SARS-CoV-2 tests for South America

**Byron Freire-Paspuel**[1], **Diana Morales-Jadan**[1], **Marlon Zambrano-Mila**[1], **Franklin Perez**[2], **Miguel Angel Garcia-Bereguiain**[1]*

**1** One Health Research Group, Universidad de Las Americas, Quito, Ecuador, **2** OneLabt, Santa Elena, Ecuador

* magbereguiain@gmail.com

## Abstract

### Background

Dozens of commercial RT-qPCR kits for SARS-CoV-2 detection are available with or without Emergency Use Authorization (EUA) by FDA or other regulatory agencies.

### Objective

We evaluated the clinical performance of two SARS-CoV-2 RT-PCR kits designed and produced in South America, "COVID-19 RT-PCR Real TM FAST (CY5)" (ATGen, Uruguay) and "ECUGEN SARS-CoV-2 RT-qPCR" (UDLA-STARNEWCORP, Ecuador), for RT-qPCR SARS-CoV2 detection using "TaqMan 2019-nCoV Assay Kit v1" (Thermofisher, USA) as a gold standard technique.

### Results

We report a great clinical performance and analytical sensitivity for the two South American kits with sensitivity values of 96.4 and 100%, specificity of 100% and limit of detection in the range of 10 copies/uL of RNA extraction.

### Conclusions

"COVID-19 RT-PCR Real TM FAST (CY5)" and "ECUGEN SARS-CoV-2 RT-qPCR" kits are reliable SARS-CoV-2 tests made in South America that have been extensively used in Uruguay, Argentina, Brazil, Bolivia and Ecuador. These locally produced SARS-CoV-2 tests have contributed to overcome supply shortages and reduce diagnosis cost, while maintaining the high quality standards of FDA EUA commercially available kits. This approach could

**Data Availability Statement:** All relevant data are within the manuscript and its Supporting Information files.

**Funding:** The authors received no specific funding for this work.

**Competing interests:** I have read the journal's policy and the authors of this manuscript have the following competing interests: The authors of this study are employees of "Universidad de Las Américas". This institution is enrolled in the commercialization "ECUGEN SARS-CoV-2 RT-PCR kit", one of the kits evaluated on this study.

be extended for other diagnostic products to improve infectious diseases surveillance at middle and low income countries beyond COVID-19 pandemic.

## Author summary

COVID-19 pandemic is the worst public health crisis that humanity has faced in the last decades. To success controlling the virus spread an unprecedented amount on molecular tests based on the technique called PCR has been necessary. To carry out viral infection tests, commercial kits are normally used by clinical laboratories. Those commercial kits are mainly produced in industrialized countries and that means a disadvantage in the access to COVID-19 testing in developing countries. Here we present the results of the evaluation of two commercial kits produced in South America for local stakeholders, showing how good quality biotech products can help to fight COVID-19 pandemic in low and middle income countries.

## Introduction

The "coronaviruses disease 2019" (COVID-19) pandemic, caused by Severe Acute Respiratory Syndrome Coronavirus 2 (SARS-CoV-2), has challenged public health systems worldwide since the initial outbreak in the Chinese city of Wuhan in December 2019 [1,2]. While SARS-CoV-2 vaccination programs progress successfully at high income countries and COVID-19 pandemic is somehow under control, the challenge persists for middle and low income countries in need to improve SARS-CoV-2 testing [3–5]. A wide variety of commercial SARS-CoV-2 RT-qPCR detection kits are available in South America for clinical use, some of them with Emergency Use Authorization (EUA) from the U.S. Food & Drug Administration (FDA) or other international recognized agencies, while others even lack of their country of origin EUA and compromise a reliable diagnosis in these high burden COVID-19 settings [6–13]. Among the SARS-CoV-2 RT-qPCR commercial kits available, "TaqMan 2019-nCoV Assay Kit v1" (Thermofisher, USA) holds FDA EUA and it is considered one of the most reliable SARS-CoV-2 RT-qPCR kits [6]. All those commercial kits are based in SARS-CoV-2 detection by targeting different genes like E, S, N or the orf1ab region.

The CDC designed FDA EUA 2019-nCoV CDC kit (IDT, USA) is based on N1 and N2 gene targets to detect SARS-CoV-2 and RNase P as an RNA extraction quality control, it is considered a gold standard worldwide for SARS-CoV-2 RT-PCR detection [14–16]. The main limitation for the CDC kit is the need to run three RT-qPCR reactions per sample. To solve this inconvenience, inspired on this CDC protocol, two SARS-CoV-2 RT-qPCR multiplex assays have been designed, produced and commercialized in South America. "COVID-19 RT-PCR Real TM FAST (CY5)" (ATGen, Uruguay) is a duplex assay including N1 viral target and RNase P as an RNA extraction quality control. "ECUGEN SARS-CoV-2 RT-qPCR k" (UDLA-STARNEWCORP, Ecuador) is a triplex assay including N1 and N2 viral targets and also RNase P as an RNA extraction quality control. So, for both South American kits, a single PCR reaction is required for SARS-CoV-2 detection, reducing the time and cost of the diagnosis.

We herein present an analytical sensitivity and clinical performance evaluation of "COVID-19 RT-PCR Real TM FAST (CY5)" (ATGen, Uruguay) and "ECUGEN SARS-CoV-2 RT-qPCR" (UDLA-STARNEWCORP, Ecuador) for SARS-CoV-2 RT-qPCR detection from

nasopharyngeal samples using "TaqMan 2019-nCoV Assay Kit v1" (Thermofisher, USA) as a gold standard technique.

## Materials and methods

### Ethics statement

This study is part of a project approved by the IRB from the Dirección Nacional de Inteligencia de la Salud (Ministerio de Salud Publica, Ecuador) under the code 008–2020.

### Study design

119 samples were included in this study. Those samples were leftovers of the RNA extractions of clinical specimens previously processed for SARS-CoV-2 test and stored at -80 C. Those RNA extractions were obtained from nasopharyngeal swabs collected on 0.5mL TE pH 8 buffer from community dwelling individuals attending Universidad de Las Américas for SARS-CoV-2 testing during November and December 2020. Also, negative controls (TE pH 8 buffer) were included as control for carryover contamination, one for each set of RNA extractions. The overall protocol for sample collection and SARS-CoV-2 diagnosis at our laboratory has been previously published [17–20].

All the clinical samples were processed with the same RNA extraction kit "AccuPrep Viral RNA extraction kit" (Bioneer, South Korea) and used for SARS-CoV-2 detection. Only the viral RNA extractions stored at -80 C were used for the three RT-qPCR protocols included in this study. For the RNA extractions, 200μL of TE pH8 buffer that contained the sample were used. At the end of the RNA extraction, RNA was eluted in 40μL elution buffer.

A total of 109 RNA samples were evaluated using ECUGEN SARS-CoV-2 RT-qPCR Kit (UDLA-STARNEWCORP, Ecuador) and a total of 80 RNA samples were evaluated using COVID-19 RT-PCR Real TM FAST (CY5) (ATGen, Uruguay). All the RNA samples of both sets were additionally processed with TaqMan 2019-nCoV Assay Kit v1 as gold standard SARS-CoV-2 detection method.

**RT-qPCR for SARS-CoV-2 detection using TaqMan 2019-nCoV Assay Kit v1.** All samples were tested following the manufacturer instructions using TaqPath 1-Step Master Mix GC in a reduced reaction volume of 15μL including 4μL of RNA sample. TaqMan 2019-nCoV Control Kit v1 was used as reaction positive control. Following the manufacturer's manual, a viral target with $Ct < 37$ is considered positive and with $37 \leq Ct < 40$ is considered inconclusive. A new PCR reaction was run for inconclusive results, and a $Ct < 40$ is sufficient to considered that run as positive. Samples that showed positive results for at least two of the three SARS-CoV-2 genes (S, N and ORF1ab) and inconclusive samples that showed recurrent positive results for at least one viral gene target were marked as SARS-CoV-2 positive. RT-qPCR assays were performed in a CFX96 Real-Time PCR Detection System (Bio-Rad).

**RT-qPCR for SARS-CoV-2 detection using ECUGEN SARS-CoV-2 RT-qPCR Kit.** Samples were tested following the manufacturer instructions in a reaction volume of 15μL including 4μL of RNA sample. The criteria for positivity were $Ct \leq 40$ for N1 and N2 targets simultaneously. Also, inconclusive samples where either N1 and N2 were positive, were repeated and if either N1 or N2 target amplified, the sample is considered positive. RT qPCR assays were performed in a CFX96 Real-Time PCR Detection System (Bio-Rad).

**RT-qPCR for SARS-CoV-2 detection using COVID-19 RT-PCR Real TM FAST (CY5) (ATGen) kit.** Samples were tested following the manufacturer instructions in a reduced reaction volume of 15μL including 3,75μL of RNA sample. Samples that presented a $Ct < 35$ for the viral target gene were considered positive, and samples with values of $35 \leq Ct \leq 40$ were

considered inconclusive. The latter were repeated and consistent results with the first test was sufficient to be considered positive. RT-qPCR assays were performed in a CFX96 Real-Time PCR Detection System (Bio-Rad).

**Analytical sensitivity.** Limit of detection (LoD) was performed using the commercially available 2019-nCoV N positive control (IDT, USA); provided at 200,000 genome equivalents/uL, it was used for calibration curves to obtain the viral loads of the samples. Viral loads can be expressed as copies/uL of RNA extraction or copies/mL of sample; the conversion factor is 200, as 0.2mL of sample is used for RNA extraction and 40uL is used as final elution volume of RNA extraction.

*Statistics analysis.* All data was analyzed in Excel and statistics were done using SPSS software.

## Results

### Clinical performance and analytical sensitivity of ECUGEN SARS-CoV-2 RT-qPCR Kit

109 samples were tested for SARS-CoV-2 with ECUGEN SARS-CoV-2 RT-qPCR Kit using TaqMan 2019-nCoV Assay Kit v1 as a gold standard. For the TaqMan 2019-nCoV Assay Kit v1, 55 samples tested positive, and 54 samples tested negative (Table 1 and S1 Data). 51 out of 54 samples tested negative for the TaqMan 2019-nCoV Assay Kit v1 were also SARS-CoV-2 negative for ECUGEN SARS-CoV-2 RT-qPCR Kit, so the specificity obtained in this study was 94.4% (95% CI = 84.6 to 98.8%). The three "false positive" samples had Ct values > 35 and viral loads of 3.1, 4.72 and 6.84 copies/uL of RNA extraction (Samples 62, 64 and 66 at S1 Data).

For the 55 SARS-CoV-2 positive samples for the TaqMan 2019-nCoV Assay Kit v1, 55 samples tested also positive for ECUGEN SARS-CoV-2 RT-qPCR Kit, resulting a sensitivity of 100.0% (95% CI = 93.51 to 100.00%) (Table 1 and S1 Data). Cohen's κ was run and almost perfect agreement between results was obtained with both kits (κ = 0.945, p<0.001). In Fig 1A, the distribution of Ct values of N gene target for SARS-CoV-2 positive samples included in the study for ECUGEN SARS-CoV-2 RT-qPCR Kit and TaqMan 2019-nCoV Assay Kit v1 is show. In Fig 1B the linear regression analysis for Ct values for N gene target for both kits is presented, yielding a $R^2$ = 0.9906.

The limit of detection (LoD) is defined as the lowest viral load in which all replicates are detected (100% sensitivity). As it is detailed in Table 2, after running 15 replicates for viral loads in the range from 500 to 2000 copies/mL, we could set the LoD for ECUGEN SARS-CoV-2 RT-qPCR Kit in 2000 copies/mL of sample (10 copies/μL of RNA extraction). Although N2 target fails for 1 out of 15 replicates at that viral load, as the criteria of positivity only requires the amplification of one of the viral target on the replicate, the LoD was the lowest viral load for 100% sensitivity for N1 target.

**Table 1. Clinical performance of ECUGEN SARS-CoV-2 RT-qPCR and COVID-19 RT-PCR Real TM FAST (CY5) kits using TaqMan 2019-nCoV Assay Kit v1 as reference methodology (% values: sensitivity).** Only SARS-CoV-2 positive samples included on the study are detailed.

| SARS-CoV-2 RT-qPCR kit | Positive Samples | False Negative Samples | Total SARS-CoV-2 positive samples |
|---|---|---|---|
| ECUGEN SARS-CoV-2 RT-qPCR Kit | 55 (100.0%) | 0 | 55 |
| COVID-19 RT-PCR Real TM FAST (CY5) | 53 (96.36%) | 2 | 55 |

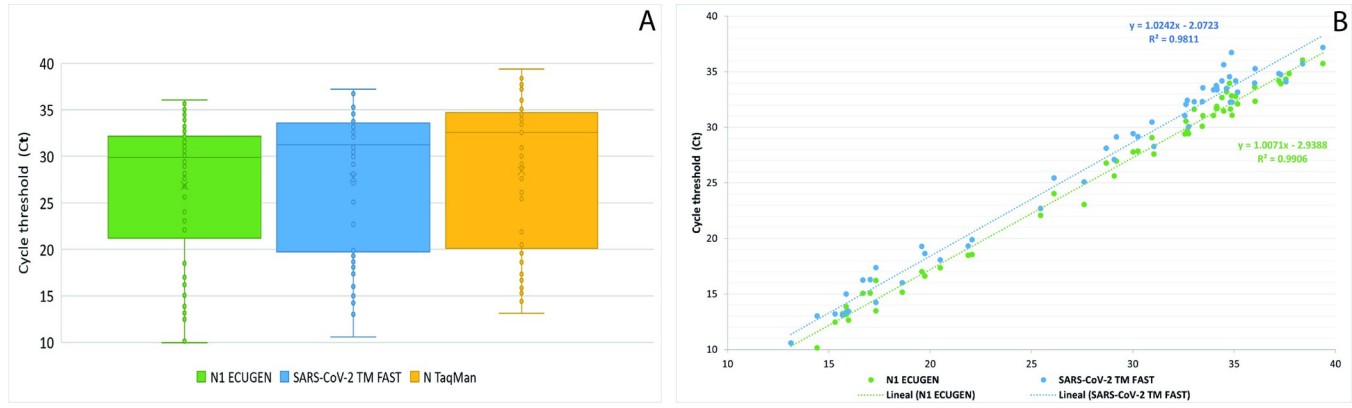

**Fig 1.** A: Cycle threshold (Ct) distribution for N gene target for SARS-CoV-2 positive samples with the three commercial kits included in this study: ECUGEN SARS-CoV-2 RT-qPCR Kit (green), COVID-19 RT-PCR Real TM FAST (blue) and TaqMan 2019-nCoV Assay Kit v1 (yellow). B: Linear regression for the Ct values for N gene target for ECUGEN SARS-CoV-2 RT-qPCR Kit (green) and COVID-19 RT-PCR Real TM FAST (blue) compared to TaqMan 2019-nCoV Assay Kit v1.

## Clinical performance and analytical sensitivity of COVID-19 RT-PCR Real TM FAST (CY5) kit

80 samples were tested for SARS-CoV-2 with of COVID-19 RT-PCR Real TM FAST (CY5) (ATGen) kit using TaqMan 2019-nCoV Assay Kit v1 as gold standard. For the TaqMan 2019-nCoV Assay Kit v1, 55 samples tested positive, and 25 samples tested negative (Table 1 and S1 Data). 24 out of 25 samples tested negative for the TaqMan 2019-nCoV Assay Kit v1 were also SARS-CoV-2 negative for COVID-19 RT-PCR Real TM FAST Kit, so the specificity obtained in this study was 96.00% (95% CI = 79.6 to 99.9%). The sample 62 (S1 Data) had a Ct value of 34.73 for N target and also presented a positive result with ECUGEN SARS-CoV-2 RT-qPCR Kit, however, this sample tested negative with the TaqMan kit.

From the 55 samples that tested positive with TaqMan 2019-nCoV Assay Kit v1, 53 samples were positive for COVID-19 RT-PCR Real TM FAST Kit, resulting a sensitivity of 96.4% (95% CI = 87.5 to 99.6%) (Table 1 and S1 Data). Samples 59 and 68 amplified for at least one SARS-CoV-2 target in different TaqMan RT-qPCR reactions, therefore, these samples were marked as positive. Cohen's κ was run and almost perfect agreement between results obtained with both kits was found (κ = 0.914, p<0.001). In Fig 1A, the distribution of Ct values of N gene target for SARS-CoV-2 positive samples included in the study for COVID-19 RT-PCR Real TM FAST Kit and TaqMan 2019-nCoV Assay Kit v1 is show. In Fig 1B the linear regression analysis for Ct values for N gene target for both kits is presented, yielding a $R^2$ = 0.9811.

As the limit of detection (LoD) is defined as the lowest viral load in which all replicates are detected (100% sensitivity), our data indicates that the LoD for COVID-19 RT-PCR Real TM FAST (CY5) Kit should be in the range of 5–10 copies/µL of RNA extraction (1000–2000 viral

**Table 2. Analytical sensitivity for ECUGEN SARS-CoV-2 RT-qPCR kit.** The ratio represents the number of positive replicates for each viral load related to the total number of replicates.

| Viral load (copies/mL) | N1 replicates | N1 sensitivity | N2 replicates | N2 sensitivity |
|---|---|---|---|---|
| 2000* | 15/15 | 100% | 14/15 | 93.3% |
| 1500 | 14/15 | 93.3% | 14/15 | 93.3% |
| 1000 | 12/15 | 80% | 12/15 | 80% |
| 500 | 12/14 | 85.7% | 10/14 | 71.4% |

RNA copies/mL of sample), as the 4 samples with viral loads within that range were detected (S1 Data), and positive samples that failed for this kit were below 5 copies/uL.

## Discussion

Although the main limitation of our study is the sample size (119 specimens), our results support that "COVID-19 RT-PCR Real TM FAST (CY5) (ATGen, Uruguay) and "ECUGEN SARS-CoV-2 RT-qPCR" (UDLA-STARNEWCORP, Ecuador) kits had a great clinical performance with sensitivity values of 96.4% and and 100%, respectively. Moreover, although a reduction of specificity was found for "COVID-19 RT-PCR Real TM FAST (CY5) (96%) and "ECUGEN SARS-CoV-2 RT-qPCR" (UDLA-Starnewcorp, Ecuador)" (94.4%) kits, we believe that the 3 "false positive" samples would actually be true positives samples as the Ct values obtained indicated that those samples had really low viral loads on the threshold for detection of the gold standard method used. Actually, one of the false positive samples was positive for both "COVID-19 RT-PCR Real TM FAST (CY5) and "ECUGEN SARS-CoV-2 RT-qPCR" kits. Additionally, both kits use the same N viral targets than the CDC protocol that do not have cross reactivity with other respiratory virus [14,15], and those 3 samples were reported as positive for the regular CDC protocol used for clinical diagnosis in our laboratory [14,15]. So, the specificity for "COVID-19 RT-PCR Real TM FAST (CY5)" and "ECUGEN SARS-CoV-2 RT-qPCR" kits could be considered 100%.

We could calculate the LoD for "ECUGEN SARS-CoV-2 RT-qPCR" at a really low viral load of 10 viral copies/uL of RNA extraction (2000 viral RNA copies/mL of sample) that it is equivalent to LoDs of high quality commercial RT-qPCR SARS-CoV-2 kits. Also, a similar LoD was estimated for "COVID-19 RT-PCR Real TM FAST (CY5). Moreover, this LoD is extremely reliable for SARS-CoV-2 diagnosis considering the viral load frequency population distributions [21,22].

In the Table 3, analytical parameters and other features for "COVID-19 RT-PCR Real TM FAST (CY5) and "ECUGEN SARS-CoV-2 RT-qPCR" RT-PCR kits are summarized. Considering the great clinical performance and analytical sensitivity for those locally designed and produced SARS-CoV-2 tests, compared to a high quality commercial kit like TaqMan 2019-nCoV Assay Kit v1 (Thermo Fisher), the current study endorses the use of these kits as a reliable alternative to expensive imported commercial kits. This would potentially allow to increase SARS-CoV-2 testing capacities by two main reasons: a) SARS-CoV-2 testing cost reduction as this locally produced RT-qPCR kits are substantially cheaper than high quality imported ones (less than 10 USD per reaction); b) supplies shortages would not affect SARS-CoV-2 testing capacities as local production is guaranteed.

Finally, we point out a common feature for both South American kits evaluated in this study: they were designed and produced by a consortium between universities and private companies. "COVID-19 RT-PCR Real TM FAST (CY5)" was created by collaboration of "Universidad de La República" (public university), "Instituto Pasteur de Montevideo" (research center) and ATGen (private company); "ECUGEN SARS-CoV-2 RT-qPCR" was created by

**Table 3. Description of ECUGEN SARS-CoV-2 RT-qPCR (UDLA-Starnewcorp, Ecuador) and COVID-19 RT-PCR Real TM FAST (CY5) (ATGen, Uruguay) features (LoD means limit of detection in copies/uL of RNA extraction elution).**

| SARS-CoV-2 RT-PCR kit | Gene Targets | Estimated LoD (copies/uL) | countries of distribution |
|---|---|---|---|
| ECUGEN SARS-CoV-2 RT-qPCR (Ecuador) | N1, N2 (virals) RNaseP (control) | 10 | Ecuador |
| COVID-19 RT-PCR Real TM FAST (CY5) (Uruguay) | N (viral) RNaseP (control) | 5–10 | Uruguay, Argentina, Bolivia, Brasil, Ecuador. |

collaboration of "Universidad de Las Américas" (private university) and STARNEWCORP (private company). In both cases, the role of the Academia has been crucial to improve good quality SARS-CoV-2 testing, as it has been suggested even for USA [23]. Moreover, we describe two cases of cross talk and knowledge transference among the Academia and the private sector, common on high incomes countries but not as usual in the context of South America. We hope that these two successful stories will inspire similar biotechnological developments in the future to improve South American public health systems and reduce the regional overall technological dependency beyond COVID-19 pandemic.

## Supporting information

**S1 Data. Ct values for all samples included in this study for all the gene targets included in the three commercial SARS-CoV-2 RT-qPCR kits tested.**
(XLSX)

## Acknowledgments

We thank Dr Tannya Lozada from "Dirección General de Investigación de la Universidad de Las Américas", and also the authorities from "Universidad de Las Américas", for logistic support to make SARS-CoV-2 diagnosis possible at our laboratories. We also thank Jose Tato and Adriana Tobon from ATGen company for their logistic support to provide ATGen kits for this study.

## Author Contributions

**Conceptualization:** Franklin Perez, Miguel Angel Garcia-Bereguiain.

**Data curation:** Byron Freire-Paspuel, Diana Morales-Jadan, Marlon Zambrano-Mila, Miguel Angel Garcia-Bereguiain.

**Formal analysis:** Byron Freire-Paspuel, Diana Morales-Jadan, Marlon Zambrano-Mila, Miguel Angel Garcia-Bereguiain.

**Funding acquisition:** Miguel Angel Garcia-Bereguiain.

**Investigation:** Byron Freire-Paspuel, Diana Morales-Jadan, Marlon Zambrano-Mila, Miguel Angel Garcia-Bereguiain.

**Methodology:** Byron Freire-Paspuel, Diana Morales-Jadan, Marlon Zambrano-Mila, Franklin Perez, Miguel Angel Garcia-Bereguiain.

**Project administration:** Byron Freire-Paspuel, Miguel Angel Garcia-Bereguiain.

**Resources:** Franklin Perez, Miguel Angel Garcia-Bereguiain.

**Supervision:** Miguel Angel Garcia-Bereguiain.

**Validation:** Miguel Angel Garcia-Bereguiain.

**Visualization:** Miguel Angel Garcia-Bereguiain.

**Writing – original draft:** Byron Freire-Paspuel, Diana Morales-Jadan, Miguel Angel Garcia-Bereguiain.

**Writing – review & editing:** Byron Freire-Paspuel, Diana Morales-Jadan, Marlon Zambrano-Mila, Franklin Perez, Miguel Angel Garcia-Bereguiain.

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
