## [Decision Letter · Decision Letter 0]

21 Jul 2021

Dear Dr Garcia-Bereguiain,

Thank you very much for submitting your manuscript "Analytical sensitivity and clinical performance of "COVID-19 RT-PCR Real TM FAST (CY5) (ATGen, Uruguay) and "ECUGEN SARS-CoV-2 RT-qPCR" (UDLA-Starnewcorp, Ecuador)": high quality-low cost local SARS-CoV-2 tests for South America." for consideration at PLOS Neglected Tropical Diseases. As with all papers reviewed by the journal, your manuscript was reviewed by members of the editorial board and by several independent reviewers. In light of the reviews (below this email), we would like to invite the resubmission of a significantly-revised version that takes into account the reviewers' comments. 

We cannot make any decision about publication until we have seen the revised manuscript and your response to the reviewers' comments. Your revised manuscript is also likely to be sent to reviewers for further evaluation.

Sincerely,

Susanna Kar Pui Lau, M.D.

Deputy Editor

Susanna Kar Pui Lau

Deputy Editor

Reviewer's Responses to Questions

**Key Review Criteria Required for Acceptance?**

**Methods**

-Are the objectives of the study clearly articulated with a clear testable hypothesis stated?

-Is the study design appropriate to address the stated objectives?

-Is the population clearly described and appropriate for the hypothesis being tested?

-Is the sample size sufficient to ensure adequate power to address the hypothesis being tested?

-Were correct statistical analysis used to support conclusions?

-Are there concerns about ethical or regulatory requirements being met?

Reviewer #1: I have problems with the limit of detection (LOD) calculations made by the authors. It appears that no replicates of the samples were tested to determine the LOD. I would expect a minimum of 4 (but ideally more) for a statistically robust analysis.

More information on the 119 clinical samples is required. When were they collected? From whom? Had they already been tested? How were they stored?

Why does the number of samples tested per kit differ (i.e. 109 for the ECUGEN and 80 for the ATGen kit)? 

Page 5: line 87. What do you mean by “reduced reaction volume”? Did you reduce the volume? If so, why?

Page 5 line 90-91 and 102 to 103. Please rewrite this sentence for clarity. What do you mean by “consistent results with the first test were considered positive”?

Page 5 line 97. How was the criteria for positivity (i.e. Ct ≤40) calculated? Please provide details.

Please identify the statistical programmes used for analyses.

Reviewer #2: The methods used in this study are the methods described by the different kits' manufacters assessed in this study. 

Nevertheless one point was unclear to me : how was the Positive / inconclusive Cycle Threshold set...this seemed a little arbitrary and could change drastically the conclusions (i.e. samples positives and negatives) . It would strengthen the study if the authors could explain how they chose CT37 for Taq man, Ct 40 for Ecugen and Ct 35 for Tim Fast

Moreover in the case of the Tim Fast kit only 3.75uL of samples was used (instead of 4uL fort the other kits...this also needs explanation )

**Results**

-Does the analysis presented match the analysis plan?

-Are the results clearly and completely presented?

-Are the figures (Tables, Images) of sufficient quality for clarity?

Reviewer #1: (No Response)

Reviewer #2: The results are clearly presented and the "raw" data are in supplemental material. Nevertheless for the samples where a duplicate experiment was necessary (for exemple samples 59 and 68 ) to draw a conclusion on the samples it would be nice to show the result of each replicate. When reading that supplemental figure it was not clear why 59 and 68 were called positive whereas 85 and 98 were called negative (Taq Man kit)

**Conclusions**

-Are the conclusions supported by the data presented?

-Are the limitations of analysis clearly described?

-Do the authors discuss how these data can be helpful to advance our understanding of the topic under study?

-Is public health relevance addressed?

Reviewer #1: (No Response)

Reviewer #2: The conclusions of this manuscript are fully supported by the datas presented here . This study is of general interest as it shows that alternative methods for SARS-COV-2 detection are as performant as the CDC approved ones and can be use for general testing.The number of samples (#100 ) is good enough to be able to draw solid conclusions.

**Editorial and Data Presentation Modifications?**

Reviewer #1: The English is sufficient but could be improved. 

A brief description of the SARS-CoV-2 genome describing all of the target genes would be helpful. 

Last paragraph can be removed Page 9 line 182-193 as I do not think it is relevant to the study.

Reviewer #2: (No Response)

**Summary and General Comments**

Reviewer #1: In their manuscript the authors have evaluated the performance of two commercial diagnostic RT-qPCR kits for the detection of SARS-CoV-2. Evaluation of kits is important but it must be done in a systematic and statistically robust way. Accordingly, this manuscript requires revision.

Reviewer #2: This study compares the efficacity of 2 novel commercial kits to the CDC approved one for SARS-Cov-2 detection . This study is straight forward and well performed. It is important for each country to be able to implement wide testing with its own ressources to avoid problems of shortage. 

The datas are presented clearly and the conclusions are strongly supported by the results.

PLOS authors have the option to publish the peer review history of their article (what does this mean?). If published, this will include your full peer review and any attached files.

Reviewer #1: Yes: William G. Dundon

Reviewer #2: No
---

## [Decision Letter · Decision Letter 1]

11 Oct 2021

Dear Dr Garcia-Bereguiain,

Thank you very much for submitting your manuscript "Analytical sensitivity and clinical performance of "COVID-19 RT-PCR Real TM FAST (CY5) (ATGen, Uruguay) and "ECUGEN SARS-CoV-2 RT-qPCR" (UDLA-Starnewcorp, Ecuador)": high quality-low cost local SARS-CoV-2 tests for South America." for consideration at PLOS Neglected Tropical Diseases. As with all papers reviewed by the journal, your manuscript was reviewed by members of the editorial board and by several independent reviewers. The reviewers appreciated the attention to an important topic. Based on the reviews, we are likely to accept this manuscript for publication, providing that you modify the manuscript according to the review recommendations. 

Sincerely,

Susanna Kar Pui Lau, M.D.

Deputy Editor

Susanna Kar Pui Lau

Deputy Editor

Reviewer's Responses to Questions

**Key Review Criteria Required for Acceptance?**

**Methods**

-Are the objectives of the study clearly articulated with a clear testable hypothesis stated?

-Is the study design appropriate to address the stated objectives?

-Is the population clearly described and appropriate for the hypothesis being tested?

-Is the sample size sufficient to ensure adequate power to address the hypothesis being tested?

-Were correct statistical analysis used to support conclusions?

-Are there concerns about ethical or regulatory requirements being met?

Reviewer #1: (No Response)

Reviewer #2: no more comments. The authors addressed my comments from the first round of review

I just found a typo page 2 line 24 "with our" ???

Reviewer #3: Straightforward methodology with manufacturer provided protocols. Details have been provided accordingly as previous reviewer's suggestion. The sample limitation seems to be a problem but unsolvable. The authors have addressed this concern in the manuscript.

**Results**

-Does the analysis presented match the analysis plan?

-Are the results clearly and completely presented?

-Are the figures (Tables, Images) of sufficient quality for clarity?

Reviewer #1: (No Response)

Reviewer #2: yes

Reviewer #3: The supplementary data is weird. For sample ID 20414, there are Ct values for TaqMan 2019-nCoV Assay Kit v1 but no result indication for this sample. Is this sample tested or not? 

The LOD range calculation is a pure deduction from the clinical samples. It is not wrong but the evidence is relatively weak.

**Conclusions**

-Are the conclusions supported by the data presented?

-Are the limitations of analysis clearly described?

-Do the authors discuss how these data can be helpful to advance our understanding of the topic under study?

-Is public health relevance addressed?

Reviewer #1: (No Response)

Reviewer #2: yes

Reviewer #3: The aim of this manuscript fits the Plos NTD in terms of focusing the low income countries and their respective situation of molecular diagnosis of COVID-19. The evaluation of these local produced kits may be of interest to relevant countries which have difficulties in accessing the golden commercial kits. However, the need of molecular diagnosis is necessary to global prevention of COVID-19. Such simple and direct testing may provide alternatives for those countries' public health systems.

**Editorial and Data Presentation Modifications?**

Reviewer #1: (No Response)

Reviewer #2: (No Response)

Reviewer #3: (No Response)

**Summary and General Comments**

Reviewer #1: The authors have satisfactorily answered all of the questions put to them. Unfortunately it seems that some of the shortcomings of their study highlighted by this reviewer were due to a lack of ATGen kits for a complete and robust evaluation. Even though this is not the fault of the authors, it still does not justify the publication of an incomplete evaluation study. I therefore recommend that the authors revise their manuscript and concentrate on the evaluation of the ECUGEN kit alone.

Reviewer #2: (No Response)

Reviewer #3: A straightforward and simple diagnostic kit evaluation from respective company. This finding can provide alternatives for low income countries to support their need of COVID-19 detection kits. In order to better commercialize the kits, a larger sampling size is definitely needed for displaying confident kit performance. Yet, this paper can still provide preliminary performance quality for the public section to consider.

PLOS authors have the option to publish the peer review history of their article (what does this mean?). If published, this will include your full peer review and any attached files.

Reviewer #1: No

Reviewer #2: No

Reviewer #3: No

Figure Files:

Data Requirements:

Reproducibility:

References

---

## [Decision Letter · Decision Letter 2]

7 Dec 2021

Dear Dr Garcia-Bereguiain,

We are pleased to inform you that your manuscript 'Analytical sensitivity and clinical performance of "COVID-19 RT-PCR Real TM FAST (CY5) (ATGen, Uruguay) and "ECUGEN SARS-CoV-2 RT-qPCR" (UDLA-STARNEWCORP, Ecuador)": high quality-low cost local SARS-CoV-2 tests for South America.' has been provisionally accepted for publication in PLOS Neglected Tropical Diseases.

Best regards,

Susanna Kar Pui Lau, M.D.

Deputy Editor

Susanna Kar Pui Lau

Deputy Editor

Reviewer's Responses to Questions

**Key Review Criteria Required for Acceptance?**

**Methods**

-Are the objectives of the study clearly articulated with a clear testable hypothesis stated?

-Is the study design appropriate to address the stated objectives?

-Is the population clearly described and appropriate for the hypothesis being tested?

-Is the sample size sufficient to ensure adequate power to address the hypothesis being tested?

-Were correct statistical analysis used to support conclusions?

-Are there concerns about ethical or regulatory requirements being met?

Reviewer #1: (No Response)

Reviewer #3: This part has been addressed by the authors.

**Results**

-Does the analysis presented match the analysis plan?

-Are the results clearly and completely presented?

-Are the figures (Tables, Images) of sufficient quality for clarity?

Reviewer #1: (No Response)

Reviewer #3: The supp. result has been appropriately modified.

**Conclusions**

-Are the conclusions supported by the data presented?

-Are the limitations of analysis clearly described?

-Do the authors discuss how these data can be helpful to advance our understanding of the topic under study?

-Is public health relevance addressed?

Reviewer #1: (No Response)

Reviewer #3: Appropriate conclusion.

**Editorial and Data Presentation Modifications?**

Reviewer #1: Language revisions are recommended to improve clarity.

Reviewer #3: Nil

**Summary and General Comments**

Reviewer #1: The authors have understood the limitations of their study and have tried to resolve them. Nevertheless, limitations still exist in the number of samples tested and the lack of standard LOD calculations for one of the kits.

Reviewer #3: The question is addressed with further elaboration on the performance of ECUGEN kit. Limitation is appropriately stated in the content.

PLOS authors have the option to publish the peer review history of their article (what does this mean?). If published, this will include your full peer review and any attached files.

Reviewer #1: No

Reviewer #3: No

---

## [Editor Report · Acceptance letter]

8 Apr 2022

Dear Dr Garcia-Bereguiain,

We are delighted to inform you that your manuscript, "Analytical sensitivity and clinical performance of "COVID-19 RT-PCR Real TM FAST (CY5) (ATGen, Uruguay) and "ECUGEN SARS-CoV-2 RT-qPCR" (UDLA-STARNEWCORP, Ecuador)": high quality-low cost local SARS-CoV-2 tests for South America," has been formally accepted for publication in PLOS Neglected Tropical Diseases.

Best regards,

Shaden Kamhawi

co-Editor-in-Chief

Paul Brindley

co-Editor-in-Chief
